# Synthesis and In Vitro Antibacterial Activity of Quaternized 10-Methoxycanthin-6-one Derivatives

**DOI:** 10.3390/molecules24081553

**Published:** 2019-04-19

**Authors:** Na Li, Dan Liu, Jiang-Kun Dai, Jin-Yi Wang, Jun-Ru Wang

**Affiliations:** 1College of Veterinary Medicine, Northwest A&F University, Yangling 712100, Shaanxi, China; lnuk@nwsuaf.edu.cn; 2College of Chemistry and Pharmacy, Northwest A&F University, Yangling 712100, Shaanxi, China; ld127222@126.com (D.L.); daijkun@hotmail.com (J.-K.D.)

**Keywords:** 10-methoxycanthin-6-one, quaternization, antibacterial, SARs

## Abstract

Background: Based on our previous work, we found that 10-methoxycanthin-6-one displayed potential antibacterial activity and quaternization was an available method for increasing the antibacterial activity. Here, we explored the antibacterial activity of quaternized 10-methoxy canthin-6-one derivatives. Methods and Results: Twenty-two new 3-*N*-benzylated 10-methoxy canthin-6-ones were designed and synthesized through quaternization reaction. The in vitro antibacterial activity against three bacteria was evaluated by the double dilution method. Moreover, the structure–activity relationships (SARs) were carefully summarized in order to guide the development of antibacterial canthin-6-one agents. Two highly active compounds (**6p** and **6t**) displayed 8-fold superiority (MIC = 3.91 µg/mL) against agricultural pathogenic bacteria *R. solanacearum* and *P. syringae* compared to agrochemical streptomycin sulfate, and showed potential activity against *B. cereus*. Moreover, these two compounds exhibited good “drug-like” properties, low cytotoxicity, and no inhibition on seed germination. Conclusions: This work provides two new effective quaternized canthin-6-one derivatives as candidate bactericide, promoting the development of natural-sourced bactericides and preservatives.

## 1. Introduction

Healthcare systems, farming, and the food production industry are the main sectors driving antibiotic consumption [1]. To maintain agricultural production and food safety, there are many circumstances where antibiotics are required in the farming and food industry [2,3]. *Bacillus cereus* is pathogenic and is the common cause of spoilage of milk, cream and cooked rice. When people eat *B. cereus* contaminated food, they will suffer from food poisoning with these symptoms’ nausea, vomiting and diarrhea [4]. The synthetic preservatives could fully retain food nutrients. However, consumers are increasingly concerned about the safety of the pure synthetic chemicals [5]. Therefore, the development of effective natural-sourced antibacterial agents for food preservation and food safety is in great demand [6]. *Ralstonia solanacearum* and *Pseudomonas syringae*, plant pathogens, could infect the plant tissues during the whole plant growth cycle [7]. Although chemical pesticides work quickly, they cause serious environmental pollution, human and animal poisoning, and pesticide residues [8]. Recently, structural modifications of natural products for the discovery of agrochemical candidates have received much attention because of their low toxicity, low residue and environmental friendliness [8,9]. Canthin-6-one 1, mainly from the Rutaceae and Simaroubaceae families, are a subclass of the tryptophan-derived *β*-carboline alkaloids [10,11]. In our previous work, we found that quaternized modification could significantly improve the antibacterial activity (Figure 1) [12,13]. For example, the antibacterial activity of compounds **2** and **4** against *B. cereus* enhanced 2-fold and 8-fold, respectively. Interestingly, 10-methoxycanthin-6-one **5**, per se has good antibacterial activity [14,15]. With the aim of finding more promising antibacterial canthin-6-one analogues, we report here a simple idea and design to generate the bioactive compounds by quaternization reaction of 10-methoxycanthin-6-one **5**.

## 2. Results and Discussion

### 2.1. Chemistry

The synthetic route of target compounds **6a**–**6v** was outlined in Scheme 1. The N_b_-benzyl-5-methoxytryptamine **8** was prepared through Borch Reduction with 5-methoxytryptamine **7** as the starting material (98% yield). Then, N_b_-benzylhexahydro-10-methoxycanthin-6-one **9** was obtained by the key Pictet–Spengler reaction (83% yield). The debenzylated compound **10** (hexahydro-10-methoxycanthin-6-one) was sequentially obtained by catalytic hydrogen transfer reaction (75% yield). Finally, the natural product 10-methoxycanthin-6-one **5** was prepared by oxidation reaction (90% yield) [15]. With this compound **5** in hand, attention was focused on modifying the 3-N position. A benzyl group in this position significantly increased the activity (Figure 1). Thus, we synthesized a series of quaternized 10-methoxycanthin-6-one derivatives **6a**–**6v** (42–72% yields) with diverse substituted benzyl group which varied in electron-inducing ability and substitution position.

The purity of the compounds (>99%) has been provided using high performance liquid chromatography (HPLC), which could be found in the Appendix A. All the structures of the target compounds (**5** and **6a**–**6v**) were confirmed by NMR and high resolution MS (HRMS) spectra. In the NMR spectra of compound **6a**, the signal of the methylene group was detected around δ = 6.39 ppm and δ = 56.6 ppm, respectively, which indicated that the quaternized 10-methoxycanthin-6-one derivatives were successfully synthesized. Moreover, the charactermatic signals of the methyl group were easily detected around δ = 3.96 ppm (^1^H NMR) and δ = 54.4 ppm (^13^C NMR). In addition, the signal of [M − Br]^+^ could be found at 359.1182 Da in HRMS spectra of compound **6a** (error = 2.2 ppm), which confirmed to the theoretical value 359.1190 Da within the allowable error range (error < 5 ppm).

### 2.2. Antibacterial Activity

The antibacterial activity in vitro of compounds **5** and **6a**–**6v** were evaluated against bacteria (*B**.** cereus, R. solanacearum and P. syringae*) using the double dilution method giving the minimum inhibitory concentration values (MICs) [16]. The concentration gradient was set according to our previous work [12,15]. Broad spectrum antibiotic fosfomycin sodium and ampicillin sodium, and agrochemical bactericide streptomycin sulfate were used as the positive controls [17,18]. The antibacterial results (Table 1) revealed that most of the quaternized 10-methoxycanthin-6-one derivatives displayed good in vitro biological activity against *R. solanacearum* and *P. syringae* compared with agrochemical streptomycin sulfate. Eight compounds (**6e**, **6f**, **6i**, **6k**, **6l**, **6m**, **6p** and **6t**) displayed 4-fold superiority (MIC = 3.91 µg/mL) against *R. solanacearum* than lead compound **5** and 8-fold superiority than streptomycin sulfate. Five compounds (**6f**, **6h**, **6i**, **6p** and **6t**) displayed 2-fold superiority (MIC = 3.91 µg/mL) against *P. syringae* compared to lead compound **5** and 8-fold superiority than streptomycin sulfate. Unfortunately, quaternized modification did not improve the activity for *B. cereus*. It is worth mentioning that four compounds (**6p**, **6q**, **6r** and **6t**) still have potential antibacterial activity against *B. cereus* compared with natural canthin-6-one **1**. Overall, compounds **6p** and **6t** were considered to be the most highly active derivatives against these three bacteria.

### 2.3. Structure–Activity Relationships

Based on the antibacterial activity data, the structure–activity relationships were carefully investigated for *R. solanacearum* and *P. syringae*. Halogen anion did not affect the activity. For example, the antibacterial activity of compounds **6u** and **6v** are equal. The substituted benzyl groups are favorable for most of the quaternized derivatives. The detailed SARs of a diverse substituted benzyl group which varied in electron-inducing ability and substitution position were summarized in Table 2. In terms of halogen atom substitution, fluorine substituted benzyl groups are disadvantageous for improving the activity. Chlorine and iodine substituents may be better. For example, compound **6t** with *meta*-iodine atom substituent displayed 4-fold superiority against *P. syringae* than compound **6c** which was replaced by a fluorine atom at *meta*-position. In most instances, large sterically hindered groups such as methyl and trifluoromethyl are beneficial for activity enhancement. For example, compound **6k** displayed 8-fold superiority (MIC = 3.91 µg/mL) against *R. solanacearum* compared to positive control streptomycin sulfate, which was replaced by the trifluoromethyl group at *ortho*-position. However, cyano and methoxy groups were exceptional and complex, which need further study. Multiple substitutions showed different activity trends. Compound **6p** exhibited excellent antibacterial activity compared with agrochemical streptomycin sulfate. Interestingly, multiple fluorine substituted benzyl groups are still harmful, such as compounds **6n** and **6o**.

### 2.4. Molecular Physicochemical Properties

In 1997, Lipinski et al. published what is widely regarded as the key paper defining physicochemical and structural properties profiles for the optimal oral availability of drugs [19]. In 2003, Clarke and Delaney reported that most herbicides and fungicides also adhere to the Lipinski rule [20]. The molecular properties of compounds **5**, **6p** and **6t** are shown in Table 3. For all the preferred compounds, the logP values were lower than 5, the nON values were lower than 10, the nONNH values were lower than 10, the MW values were lower than 500 and the nrotb values were lower than 5. Their physicochemical data conform to the Lipinski rule which demonstrate they may have good “drug-like” properties. Further more, the topological polar surface area (TPSA), a parameter, defined as the surface sum over all polar atoms and used extensively in medicinal chemistry to predict absorption and optimize a compound’s membrane permeability, was also calculated. It is suggested that compounds with a TPSA greater than 140 Å^2^ tend to be poor at permeating cell membranes [21]. Based on the predicted data mentioned above, the calculated TPSA values for these three compounds would indicate a possible good bioavailability. Interestingly, the TPSA values of **6p** and **6t** are lower than **5**, which indicate that the modified molecules may pass through the bacteria cell membrane more easily.

### 2.5. In Vitro Toxicity of the Highly Active Derivatives ***6p*** and ***6t***

Many natural and synthetic agents with the ability to interact with DNA often have a relatively low therapeutic index, most likely due to the unspecific manner of these agents in causing DNA damage both in lesions and in highly proliferative normal tissues [22]. Similarly, the frequency of utilization of β-carbolines is greatly reduced because of their cytotoxic activity toward mammalian cells (low selectivity) [14,22]. We determined the cytotoxic effects of the highly bioactive quaternized derivatives 6p and 6t on BHK-21 cells (Baby hamster Syrian kidney cells). As shown in Table 4, the cell viabilities of all the tested compounds were more than 96% on BHK-21 cells even up to concentrations of 15.62 μg/mL (2-fold or 4-fold MIC values). The results suggest that these molecules are of low toxicity against mammalian cells when they act as antibacterial agents.

### 2.6. Germination Rate Evaluation of the Highly Active Derivatives ***6p*** and ***6t***

The disease caused by *R. solanacearum* could be controlled before planting [23]. Thus, the effect of bactericides on seed germination rate is more important. Compounds 6p and 6t were evaluated for allelopathic activity against dicotyledon turnip (*Raphanus sativus*) and monocotyledon wheat (*Triticum aestivum* L.) seeds by determining the germination rates with respect to the positive control, glyphosate, a broad-spectrum systemic herbicide [14]. As shown in Table 5, compounds 6p and 6t showed no inhibition on germination rates under an effective antibacterial dose (3.91 μg/mL), which indicated that these molecules would not inhibit plant growth when it acts as bactericides.

## 3. Materials and Methods

### 3.1. General Details

All the reagents and solvents were obtained locally or purified according to standard methods. Melting points were determined using a digital melting-point apparatus and were uncorrected. ^1^H NMR (500 MHz) and ^13^C NMR (125 MHz) spectra were recorded using a Bruker Avance III 500 MHz instrument (Bruker, Madison, WI, USA) with Tetramethylsilane(TMS) as the internal standard and dimethylsulfoxide (DMSO-*d*_6_) as the solvent. High-resolution mass spectroscopy (HRMS) was undertaken using an AB SCIEX Triple TOF 5600^+^ spectrometer (Carlsbad, CA, USA). The purity of all compounds was detected by HPLC (Shimadzu LC-20A, Tokyo, Japan) with the system equipped with a C18 trap column (4.6 mm i.d. × 250 mm, 5 μm). The mobile phase was an aqueous methanol solution (50%) with the flow rate of 1.0 mL/min. The silica gel and GF254 silica gel of analytical thin-layer chromatography (TLC) were produced by the Qingdao Haiyang Chemical Co., Ltd. (Qingdao, China).

### 3.2. Synthesis of Target Compounds ***6a***−***6v***

The 10-methoxy canthin-6-one **5** could be easily obtained according to our previous work [15]. Compound **5** was dissolved in CH_3_CN (50 mL) and the diverse substituted benzyl bromide or chloride (5 eq.) was added. The solution was then stirred at 80 °C until the reaction is completed. The reaction solution was concentrated under reduced pressure, and purified by flash column chromatography using chloroform/methanol (30:1, *v*/*v*) as the eluent.

*3-(2-Fluorobenzyl)-10-methoxy-6-oxo-6H-indolo[3,2,1-de][1,5]naphthyridin-3-ium bromide***6a**: yield 60%; yellow-green solid powder; m.p. 224.2–224.7 °C; ^1^H NMR (500 MHz, DMSO-*d*_6_) δ 9.29 (d, *J* = 5.0 Hz, 1H), 8.96 (d, *J* = 5.0 Hz, 1H), 8.59 (d, *J* = 10.0 Hz, 1H), 8.43 (d, *J* = 10.0 Hz, 1H), 8.25 (d, *J* = 5.0 Hz, 1H), 7.58–7.50 (m, 1H), 7.51–7.46 (m, 1H), 7.40 (d, *J* = 10.0 Hz, 1H), 7.36–7.32 (m, 2H), 7.24 (d, *J* = 10.0 Hz, 1H), 6.39 (s, 2H), 3.96 (s, 3H); ^13^C NMR (125 MHz, DMSO-*d*_6_) δ 158.7, 157.5, 142.6, 136.7, 136.3, 134.6, 134.5, 132.0, 131.9, 130.6, 130.2, 130.0, 125.7, 124.1, 122.8, 120.1, 117.9, 116.5, 116.3, 109.3, 56.6, 54.4. HRMS (ESI) *m*/*z* calcd for C_22_H_16_BrFN_2_O_2_ [M − Br]^+^ 359.1190, found 359.1182.

*3-(4-Fluorobenzyl)-10-methoxy-6-oxo-6H-indolo[3,2,1-de][1,5]naphthyridin-3-ium bromide***6b**: yield 61%; yellow-green solid powder; m.p. 233.2–235.0 °C; ^1^H NMR (500 MHz, DMSO-*d*_6_) δ 9.44 (d, *J* = 5.0 Hz, 1H), 9.03 (d, *J* = 5.0 Hz, 1H), 8.67 (d, *J* = 10.0 Hz, 1H), 8.43 (d, *J* = 5.0 Hz, 1H), 8.31 (s, 1H), 7.58−7.54 (m, 3H), 7.40 (d, *J* = 10.0 Hz, 1H), 7.32−7.28 (m, 2H), 6.35 (s, 2H), 3.98 (s, 3H); ^13^C NMR (125 MHz, DMSO-*d*_6_) δ 158.6, 157.5, 142.4, 136.5, 136.2, 134.6, 131.3, 130.6, 130.5, 130.3, 130.1, 124.1, 122.6, 120.3, 117.8, 116.5, 116.4, 109.4, 58.5, 56.6. HRMS (ESI) *m*/*z* calcd for C_22_H_16_BrFN_2_O_2_ [M − Br]^+^ 359.1190, found 359.1180.

*3-(3-Fluorobenzyl)-10-methoxy-6-oxo-6H-indolo[3,2,1-de][1,5]naphthyridin-3-ium bromide***6c**: yield 65%; brick yellow solid powder; m.p. 241.6–241.9 °C; ^1^H NMR (500 MHz, DMSO-*d*_6_) δ 9.40 (d, *J* = 5.0 Hz, 1H), 9.00 (d, *J* = 5.0 Hz, 1H), 8.64 (d, *J* = 10.0 Hz, 1H), 8.40 (d, *J* = 10.0 Hz, 1H), 8.27 (d, *J* = 2.5 Hz, 1H), 7.55–7.52 (m, 3H), 7.51 (d, *J* = 10.0 Hz, 1H), 7.39–7.25 (m, 2H), 6.31 (s, 2H), 3.95 (s, 3H); ^13^C NMR (125 MHz, DMSO-*d*_6_) δ 163.7, 161.7, 158.6, 157.5, 142.4, 136.5, 136.2, 134.6, 134.6, 130.6, 130.5, 130.3, 130.1, 124.1, 122.6, 120.2, 117.8, 116.5, 116.4, 109.3, 58.5, 56.6. HRMS (ESI) *m*/*z* calcd for C_22_H_16_BrFN_2_O_2_ [M − Br]^+^ 359.1190, found 359.1184.

*3-(3-Cyanobenzyl)-10-methoxy-6-oxo-6H-indolo[3,2,1-de][1,5]naphthyridin-3-ium chloride***6d**: yield 45%; brick red solid powder; m.p. 225.8–227.7 °C; ^1^H NMR (500 MHz, DMSO-*d*_6_) δ 9.38 (d, *J* = 10.0 Hz, 1H), 9.02 (d, *J* = 5.0 Hz, 1H), 8.53 (d, *J* = 10.0 Hz, 1H), 8.46 (d, *J* = 10.0 Hz, 1H), 8.30 (s, 1H), 7.94 (d, *J* = 10.0 Hz, 2H), 7.60 (d, *J* = 10 Hz, 1H), 7.55 (d, *J* = 5.0 Hz, 2H), 7.39 (d, J = 10.0 Hz, 1H), 6.45 (s, 2H), 3.99 (s, 3H); ^13^C NMR (125 MHz, DMSO-*d*_6_) δ 158.6, 157.5, 142.5, 136.5, 136.2, 135.1, 134.6, 134.5, 130.3, 130.1, 129.6, 129.4, 127.9, 124.1, 122.6, 120.2, 117.8, 112.6, 109.3, 59.3, 56.6. HRMS (ESI) *m*/*z* calcd for C_23_H_16_ClN_3_O_2_ [M − Cl]^+^ 366.1237, found 366.1236.

*3-(2-Chlorobenzyl)-10-methoxy-6-oxo-6H-indolo[3,2,1-de][1,5]naphthyridin-3-ium chloride***6e**: yield 44%; yellow solid powder; m.p. 231.2–232.9 °C; ^1^H NMR (500 MHz, DMSO-*d*_6_) δ 9.44 (s, 1H), 9.00 (s, 1H), 8.62 (d, *J* =10.0 Hz, 1H), 8.40 (d, *J* = 10.0 Hz, 1H), 8.26 (s, 1H), 7.54 (d, *J* = 5.0 Hz, 1H), 7.42–7.32 (m, 5H), 6.36 (s, 2H), 3.95 (s, 3H); ^13^C NMR (125 MHz, DMSO-*d*_6_) δ 158.7, 157.5, 142.6, 136.6, 136.3, 136.2, 135.1, 135.1, 134.6, 130.3, 130.1, 130.1, 129.5, 129.4, 129.3, 127.9, 122.6, 120.2, 117.8, 109.5, 59.3, 56.6. HRMS (ESI) *m*/*z* calcd for C_22_H_16_Cl_2_N_2_O_2_ [M − Cl]^+^ 375.0894, found 375.0900.

*3-(3-Chlorobenzyl)-10-methoxy-6-oxo-6H-indolo[3,2,1-de][1,5]naphthyridin-3-ium chloride***6f**: yield 47%; brick yellow solid powder; m.p. 229.5–231.2 °C; ^1^H NMR (500 MHz, DMSO-*d*_6_) δ 9.11 (d, *J* = 5.0 Hz, 1H), 8.75 (d, *J* = 5.0 Hz, 1H), 8.38 (d, *J* = 10.0 Hz, 1H), 8.30 (d, *J* = 10.0 Hz, 1H), 8.03 (s, 1H), 7.61–7.32 (m, 5H), 7.23 (d, *J* = 10.0 Hz, 1H), 6.18 (s, 2H), 3.90 (s, 3H); ^13^C NMR (125 MHz, DMSO-*d*_6_) δ 158.6, 157.5, 142.5, 136.6, 136.2, 134.7, 134.6, 134.1, 134.0, 132.4, 130.3, 130.2, 130.0, 130.0, 129.5, 124.1, 122.7, 120.2, 117.9, 109.3, 58.5, 56.6. HRMS (ESI) *m*/*z* calcd for C_22_H_16_Cl_2_N_2_O_2_ [M − Cl]^+^ 375.0894, found 375.0891.

*10-Methoxy-3-(2-methylbenzyl)-6-oxo-6H-indolo[3,2,1-de][1,5]naphthyridin-3-ium bromide***6g**: yield 68%; brick yellow solid powder; m.p. > 280 °C; ^1^H NMR (500 MHz, DMSO-*d*_6_) δ 9.18 (d, *J* = 10.0 Hz, 1H), 9.00 (d, *J* = 5.0 Hz, 1H), 8.44 (d, *J* = 10.0 Hz, 2H), 8.31 (d, *J* = 2.5 Hz, 1H), 7.58−7.56 (m, 1H), 7.37–7.34 (m, 2H), 7.28 (t, *J* = 7.5 Hz, 1H), 7.09 (t, *J* = 7.5 Hz, 1H), 6.55 (d, *J* = 10.0 Hz, 1H), 6.33 (s, 2H), 3.96 (s, 3H), 2.43 (s, 3H); ^13^C NMR (125 MHz, DMSO-*d*_6_) δ 158.6, 157.7, 142.4, 136.7, 136.3, 136.0, 134.7, 134.5, 133.8, 130.9, 130.8, 130.3, 128.9, 127.0, 126.1, 124.2, 122.7, 120.3, 117.8, 109.4, 57.8, 56.6, 19.3. HRMS (ESI) *m*/*z* calcd for C_23_H_19_BrN_2_O_2_ [M − Br]^+^ 355.1441, found 355.1439.

*10-Methoxy-3-(4-methylbenzyl)-6-oxo-6H-indolo[3,2,1-de][1,5]naphthyridin-3-ium bromide***6h**: yield 71%; yellow solid powder; m.p. 228.6–228.9 °C; ^1^H NMR (500 MHz, DMSO-*d*_6_) δ 9.40 (d, *J* = 5.0 Hz, 1H), 9.00 (d, *J* = 2.5 Hz, 1H), 8.63 (d, *J* = 10.0 Hz, 1H), 8.42 (d, *J* = 10.0 Hz, 1H), 8.27 (d, *J* = 5.0 Hz, 1H), 7.56 (d, *J* = 5.0 Hz, 1H), 7.39−7.34 (m, 3H), 7.26−7.25 (m, 2H), 6.29 (s, 2H), 3.97 (s, 3H), 2.31 (s, 3H); ^13^C NMR (125 MHz, DMSO-*d*_6_) δ 158.6, 157.5, 142.4, 139.0, 136.4, 136.2, 136.0, 134.6, 134.5, 132.1, 130.2, 130.1, 128.0, 124.1, 122.6, 120.2, 117.8, 109.3, 59.2, 56.6, 21.1. HRMS (ESI) *m*/*z* calcd for C_23_H_19_BrN_2_O_2_ [M − Br]^+^ 355.1441, found 355.1436.

*10-Methoxy-3-(3-methylbenzyl)-6-oxo-6H-indolo[3,2,1-de][1,5]naphthyridin-3-ium bromide***6i**: yield 70%; yellow solid powder; m.p. 222.4–222.7 °C, ^1^H NMR (500 MHz, DMSO-*d*_6_) δ 9.38 (d, *J* = 5.0 Hz, 1H), 8.98 (d, *J* = 10.0 Hz, 1H), 8.57 (d, *J* = 10.0 Hz, 1H), 8.35 (d, *J* = 10.0 Hz, 1H), 8.24 (d, *J* = 5.0 Hz, 1H), 7.50–7.48 (m, 1H), 7.34 (d, *J* = 10.0 Hz, 1H), 7.28 (t, *J* = 10.0 Hz, 1H), 7.19–7.17 (m, 3H), 6.27 (s, 2H), 3.92 (s, 3H), 2.25 (s, 3H); ^13^C NMR (125 MHz, DMSO-*d*_6_) δ 158.6, 157.5, 142.5, 139.0, 136.5, 136.2, 135.1, 134.6, 134.5, 130.3, 130.1, 130.0, 129.5, 128.3, 125.0, 124.1, 122.6, 120.2, 117.8, 109.3, 59.2, 56.6, 21.4. HRMS (ESI) *m*/*z* calcd for C_23_H_19_BrN_2_O_2_ [M − Br]^+^ 355.1441, found 355.1449.

*10-Methoxy-3-(3-methoxybenzyl)-6-oxo-6H-indolo[3,2,1-de][1,5]naphthyridin-3-ium chloride***6j**: yield 43%; brick yellow solid powder; m.p. 225.3–226.0 °C; ^1^H NMR (500 MHz, CD_3_OD) δ 9.11 (d, *J* = 5.0 Hz, 1H), 8.74 (d, *J* = 10.0 Hz, 1H), 8.55 (d, *J* = 10.0 Hz, 1H), 8.30 (d, *J* = 10.0 Hz, 1H), 7.95 (d, *J* = 2.5 Hz, 1H), 7.48 (d, *J* = 10.0 Hz, 2H), 7.37−7.35 (m, 1H), 7.30 (d, *J* = 10.0 Hz, 1H), 7.06 (d, *J* = 10.0 Hz, 2H), 6.20 (s, 2H), 3.98 (s, 3H), 3.83 (s, 3H); ^13^C NMR (125 MHz, CD_3_OD) δ 160.5, 158.9, 157.5, 141.3, 137.0, 136.1, 134.0, 134.0, 132.0, 129.6, 129.6, 129.0, 125.3, 123.6, 122.4, 121.6, 119.2, 117.6, 114.8, 107.8, 59.6, 55.6, 54.9. HRMS (ESI) *m*/*z* calcd for C_23_H_19_ClN_2_O_3_ [M − Cl]^+^ 371.1390, found 371.1383.

*10-Methoxy-6-oxo-3-(2-(trifluoromethyl)benzyl)-6H-indolo[3,2,1-de][1,5]naphthyridin-3-ium bromide***6k**: yield 70%; yellow solid powder; m.p. 243.6–244.1 °C; ^1^H NMR (500 MHz, DMSO-*d*_6_) δ 9.22 (d, *J* = 5.0 Hz, 1H), 9.00 (d, *J* = 10.0 Hz, 1H), 8.47 (d, *J* = 10.0 Hz, 1H), 8.30−8.28 (m, 2H), 7.96 (d, *J* = 7.5 Hz, 1H), 7.69–7.55 (m, 3H), 7.36 (d, *J* = 10.0 Hz, 1H), 6.82 (d, *J* = 5.0 Hz, 1H), 6.51 (s, 2H), 3.98 (s, 3H); ^13^C NMR (125 MHz, DMSO-*d*_6_) δ 158.7, 157.6, 143.0, 137.1, 136.4, 135.7, 134.8, 134.7, 134.0, 132.8, 130.8, 130.0, 129.8, 128.2, 127.3, 126.3, 124.1, 122.9, 120.3, 117.9, 109.4, 56.6, 56.5. HRMS (ESI) *m*/*z* calcd for C_23_H_16_BrF_3_N_2_O_2_ [M − Br]^+^ 409.1158, found 409.1157.

*10-Methoxy-6-oxo-3-(4-(trifluoromethyl)benzyl)-6H-indolo[3,2,1-de][1,5]naphthyridin-3-ium bromide***6l**: yield 68%; yellow solid powder; m.p. 238.5–238.9 °C; ^1^H NMR (500 MHz, DMSO-*d*_6_) δ 9.43 (d, *J* = 10.0 Hz, 1H), 9.01 (d, *J* = 5.0 Hz, 1H), 8.63 (d, *J* = 10.0 Hz, 1H), 8.41 (d, *J* = 10.0 Hz, 1H), 8.28 (s, 1H), 7.92 (s, 1H), 7.77 (d, *J* = 5.0 Hz, 1H), 7.64 (s, 2H), 7.54 (d, *J* = 10.0 Hz, 1H), 7.39 (d, *J* = 10.0 Hz, 1H), 6.43 (s, 2H), 3.95 (s, 3H); ^13^C NMR (125 MHz, DMSO-*d*_6_) δ 158.6, 157.6, 142.7, 139.7, 136.7, 136.2, 134.8, 134.7, 130.5, 130.0, 128.6, 126.4, 125.5, 124.1, 123.3, 122.7, 120.3, 117.9, 109.3, 58.6, 56.6. HRMS (ESI) *m*/*z* calcd for C_23_H_16_BrF_3_N_2_O_2_ [M − Br]^+^ 409.1158, found 409.1162.

*10-Methoxy-6-oxo-3-(3-(trifluoromethyl)benzyl)-6H-indolo[3,2,1-de][1,5]naphthyridin-3-ium bromide***6m**: yield 72%; brick red solid powder; m.p. 223.3–223.7 °C; ^1^H NMR (500 MHz, DMSO-*d*_6_) δ 9.38 (d, *J* = 5.0 Hz, 1H), 8.99 (d, *J* = 5.0 Hz, 1H), 8.53 (d, *J* = 10.0 Hz, 1H), 8.40 (d, *J* = 5.0 Hz, 1H), 8.26 (d, *J* = 2.5 Hz, 1H), 7.77 (d, *J* = 5.0 Hz, 2H), 7.57–7.54 (m, 3H), 7.35 (d, *J* = 10.0 Hz, 1H), 6.42 (s, 2H), 3.93 (s, 3H); ^13^C NMR (125 MHz, DMSO-*d*_6_) δ 158.6, 157.6, 142.6, 136.6, 136.3, 136.2, 134.8, 134.7, 132.0, 130.7, 130.5, 130.2, 130.0, 126.1, 125.4, 125.0, 124.1, 122.7, 120.3, 117.8, 109.4, 58.5, 56.6. HRMS (ESI) *m*/*z* calcd for C_23_H_16_BrF_3_N_2_O_2_ [M − Br]^+^ 409.1158, found 409.1165.

*3-(2,5-Difluorobenzyl)-10-methoxy-6-oxo-6H-indolo[3,2,1-de][1,5]naphthyridin-3-ium bromide***6n**: yield 61%; yellow solid powder; m.p. 245.1–245.8 °C; ^1^H NMR (500 MHz, DMSO-*d*_6_) δ 9.32 (d, *J* = 5.0 Hz, 1H), 8.99 (d, *J* = 10.0 Hz, 1H), 8.58 (d, *J* = 10.0 Hz, 1H), 8.43 (d, *J* = 5.0 Hz, 1H), 8.29 (d, *J* = 5.0 Hz, 1H), 7.58–7.56 (m, 1H), 7.50–7.33 (m, 3H), 7.21−7.19 (m, 1H), 6.39 (s, 2H), 3.95 (s, 3H); ^13^C NMR (125 MHz, DMSO-*d*_6_) δ 158.6, 157.6, 142.8, 136.8, 136.3, 134.7, 134.6, 130.7, 130.1, 124.2, 124.2, 124.1, 122.8, 120.2, 118.1, 117.9, 117.9, 116.6, 116.4, 109.4, 56.6, 53.9. HRMS (ESI) *m*/*z* calcd for C_22_H_15_BrF_2_N_2_O_2_ [M − Br]^+^ 377.1096, found 377.1103.

*3-(2,4-Difluorobenzyl)-10-methoxy-6-oxo-6H-indolo[3,2,1-de][1,5]naphthyridin-3-ium bromide***6o**: yield 58%; yellow solid powder; m.p. 230.6–231.9 °C; ^1^H NMR (500 MHz, DMSO-*d*_6_) δ 9.29 (d, *J* = 5.0 Hz, 1H), 8.98 (d, *J* = 5.0 Hz, 1H), 8.64 (d, *J* = 10.0 Hz, 1H), 8.45 (d, *J* = 10.0 Hz, 1H), 8.28 (d, *J* = 2.5 Hz, 1H), 7.60−7.54 (m, 2H), 7.48–7.42 (m, 2H), 7.21−7.18 (m, 1H), 6.38 (s, 2H), 3.98 (s, 3H); ^13^C NMR (125 MHz, DMSO-*d*_6_) δ 158.6, 157.5, 142.5, 136.7, 136.3, 134.6, 134.5, 132.0, 131.9, 131.2, 130.6, 130.0, 124.0, 122.8, 120.1, 118.7, 118.5, 117.9, 109.3, 105.2, 56.6, 53.8. HRMS (ESI) *m*/*z* calcd for C_22_H_15_BrF_2_N_2_O_2_ [M − Br]^+^ 377.1096, found 377.1099.

*3-(3,4-Dichlorobenzyl)-10-methoxy-6-oxo-6H-indolo[3,2,1-de][1,5]naphthyridin-3-ium bromide***6p**: yield 46%; orange solid powder; m.p. 223.7–224.5 °C; ^1^H NMR (500 MHz, DMSO-*d*_6_) δ 9.40 (d, *J* = 5.0 Hz, 1H), 9.02 (d, *J* = 10.0 Hz, 1H), 8.60 (d, *J* = 10.0 Hz, 1H), 8.42 (d, *J* = 10.0 Hz, 1H), 8.30 (d, *J* = 2.5 Hz, 1H), 7.77 (d, *J* = 5.0 Hz, 1H), 7.73 (d, *J* = 10.0 Hz, 1H), 7.59−7.56 (m, 1H), 7.45−7.43 (m, 1H), 7.40 (d, *J* = 10.0 Hz, 1H), 6.35 (s, 2H), 3.98 (s, 3H); ^13^C NMR (125 MHz, DMSO-*d*_6_) δ 158.7, 157.6, 142.6, 136.7, 136.2, 135.8, 134.7, 134.7, 132.2, 131.6, 130.4, 130.2, 130.0, 128.6, 128.5, 124.1, 122.7, 120.3, 117.8, 109.4, 57.9, 56.6. HRMS (ESI) *m*/*z* calcd for C_22_H_15_BrCl_2_N_2_O_2_ [M − Br]^+^ 409.0505, found 409.0504.

*3-(2-Bromobenzyl)-10-methoxy-6-oxo-6H-indolo[3,2,1-de][1,5]naphthyridin-3-ium bromide***6q**: yield 62%; brick yellow solid powder; m.p. 235.5–236.0 °C; ^1^H NMR (500 MHz, DMSO-*d*_6_) δ 9.20 (d, *J* = 10.0 Hz, 1H), 8.97 (d, *J* = 5.0 Hz, 1H), 8.47−8.45 (m, 2H), 8.28 (d, *J* = 2.5 Hz, 1H), 7.81 (d, *J* = 5.0 Hz, 1H), 7.59 (d, *J* = 10.0 Hz, 1H), 7.42–7.28 (m, 3H), 6.82–6.81 (m, 1H), 6.34 (s, 2H), 3.96 (s, 3H); ^13^C NMR (125 MHz, DMSO-*d*_6_) δ 158.7, 157.6, 142.6, 136.9, 136.4, 134.7, 134.6, 134.2, 133.6, 131.2, 130.9, 130.2, 129.1, 129.0, 124.1, 122.8, 122.4, 120.2, 117.9, 109.4, 59.6, 56.6. HRMS (ESI) *m*/*z* calcd for C_22_H_16_Br_2_N_2_O_2_ [M − Br]^+^ 419.0389, found 419.0355.

*3-(4-Bromobenzyl)-10-methoxy-6-oxo-6H-indolo[3,2,1-de][1,5]naphthyridin-3-ium bromide***6r**: yield 65%; yellow solid powder; m.p. 234.5–235.2 °C; ^1^H NMR (500 MHz, DMSO-*d*_6_) δ 9.39 (d, *J* = 10.0 Hz, 1H), 9.01 (d, *J* = 5.0 Hz, 1H), 8.59 (d, *J* = 10.0 Hz, 1H), 8.44 (d, *J* = 10.0 Hz, 1H), 8.28 (d, *J* = 2.5 Hz, 1H), 7.65 (d, *J* = 5.0 Hz, 2H), 7.59−7.57 (m, 1H), 7.41–7.38 (m, 3H), 6.32 (s, 2H), 3.98 (s, 3H); ^13^C NMR (125 MHz, DMSO-*d*_6_) δ 158.6, 157.5, 142.5, 136.6, 136.2, 134.7, 134.6, 134.4, 132.4, 130.3, 130.2, 130.0, 124.1, 122.7, 122.7, 120.2, 117.9, 109.3, 58.6, 56.6. HRMS (ESI) *m*/*z* calcd for C_22_H_16_Br_2_N_2_O_2_ [M − Br]^+^ 419.0389, found 419.0381.

*3-(3-Bromobenzyl)-10-methoxy-6-oxo-6H-indolo[3,2,1-de][1,5]naphthyridin-3-ium bromide***6s**: yield 64%; brick red solid powder; m.p. 238.7–239.1 °C; ^1^H NMR (500 MHz, DMSO-*d*_6_) δ 9.41 (d, *J* = 10.0 Hz, 1H), 9.01 (d, *J* = 5.0 Hz, 1H), 8.62 (d, *J* = 10.0 Hz, 1H), 8.43 (d, *J* = 10.0 Hz, 1H), 8.29 (d, *J* = 2.5 Hz, 1H), 7.71 (s, 1H), 7.64−7.62 (m, 1H), 7.58–7.56 (m, 1H), 7.42–7.40 (m, 3H), 6.35 (s, 2H), 3.98 (s, 3H); ^13^C NMR (125 MHz, DMSO-*d*_6_) δ 158.6, 157.6, 142.6, 137.5, 136.6, 136.2, 134.7, 134.6, 132.3, 131.7, 130.7, 130.4, 130.0, 127.0, 124.1, 122.8, 122.7, 120.2, 117.8, 109.3, 58.4, 56.6. HRMS (ESI) *m*/*z* calcd for C_22_H_16_Br_2_N_2_O_2_ [M − Br]^+^ 419.0389, found 419.0386.

*3-(3-Iodobenzyl)-10-methoxy-6-oxo-6H-indolo[3,2,1-de][1,5]naphthyridin-3-ium bromide***6t**: yield 42%; red solid powder; m.p. 245.6–246.3 °C; ^1^H NMR (500 MHz, DMSO-*d*_6_) δ 9.38 (d, *J* = 10.0 Hz, 1H), 8.99 (d, *J* = 5.0 Hz, 1H), 8.58 (d, *J* = 10.0 Hz, 1H), 8.44 (d, *J* = 5.0 Hz, 1H), 8.28 (d, *J* = 2.5 Hz, 1H), 7.81 (d, *J* = 10.0 Hz, 2H), 7.60–7.57 (m, 1H), 7.39 (d, *J* = 10.0 Hz, 1H), 7.23 (d, *J* = 5.0 Hz, 2H), 6.29 (s, 2H), 3.98 (s, 3H); ^13^C NMR (125 MHz, DMSO-*d*_6_) δ 158.7, 157.6, 142.6, 136.9, 136.4, 134.7, 134.6, 134.2, 133.6, 131.2, 130.9, 130.2, 129.1, 129.0, 124.1, 122.8, 122.4, 120.2, 117.9, 109.4, 60.2, 56.6. HRMS (ESI) *m*/*z* calcd for C_22_H_16_BrIN_2_O_2_ [M − Br]^+^ 467.0251, found 467.0251.

*3-Benzyl-10-methoxy-6-oxo-6H-indolo[3,2,1-de][1,5]naphthyridin-3-ium bromide***6u**: yield 65%; yellow solid powder; m.p. 224.1–224.5 °C; ^1^H NMR (500 MHz, DMSO-*d*_6_) δ 9.43 (d, *J* = 5.0 Hz, 1H), 9.02 (d, *J* = 10.0 Hz, 1H), 8.63 (d, *J* = 10.0 Hz, 1H), 8.40 (d, *J* = 10.0 Hz, 1H), 8.28 (d, *J* = 2.5 Hz, 1H), 7.56−7.53 (m, 1H), 7.44–7.37 (m, 6H), 6.36 (s, 2H), 3.97 (s, 3H); ^13^C NMR (125 MHz, DMSO-*d*_6_) δ 158.6, 157.5, 142.5, 136.5, 136.2, 135.1, 134.6, 134.5, 130.3, 130.1, 129.6, 129.4, 127.9, 124.1, 122.6, 120.2, 117.8, 109.3, 59.3, 56.6. HRMS (ESI) *m*/*z* calcd for C_22_H_17_BrN_2_O_2_ [M − Br]^+^ 341.1284, found 341.1275.

*3-Benzyl-10-methoxy-6-oxo-6H-indolo[3,2,1-de][1,5]naphthyridin-3-ium chloride***6v**: yield 56%; yellow solid powder; m.p. 222.4–222.9°C; ^1^H NMR (500 MHz, DMSO-*d*_6_) δ 9.42 (d, *J* = 5.0 Hz, 1H), 9.01 (d, *J* = 5.0 Hz, 1H), 8.62 (d, *J* = 10.0 Hz, 1H), 8.44 (d, *J* = 10.0 Hz, 1H), 8.29(s, 1H), 7.58 (d, *J* = 10.0 Hz, 1H), 7.44–7.38 (m, 6H), 6.35 (s, 2H), 3.98 (s, 3H); ^13^C NMR (125 MHz, DMSO-*d*_6_) δ 158.6, 157.6, 142.5, 136.5, 136.2, 135.1, 134.6, 134.6, 130.3, 130.1, 129.6, 129.4, 127.9, 124.1, 122.6, 120.2, 117.9, 109.3, 59.3, 56.6. HRMS (ESI) *m*/*z* calcd for C_22_H_17_ClN_2_O_2_ [M − Cl]^+^ 341.1284, found 341.1274.

### 3.3. Antibacterial Assay

The MIC values were determined as described by the National Committee for Clinical Laboratory Standards with minor modification [15,24,25,26]. *B. cereus* (CGMCC 1.1846) was purchased from the China General Microbiological Culture Collection Center. *R. solanacearum* and *P. syringae* were provided by the College of Plant Protection, Northwest A&F University [27]. The MIC was defined as the minimum inhibitory concentration, each compound resulting in visible inhibition on bacteria growth (incubation at 37 °C for 12–14 h). Each bacterial suspension was adjusted to a concentration of 1 × 10^5^ CFU/mL. All compounds were thoroughly dried before weighing. Initially, the compounds were dissolved in dimethyl sulfoxide (DMSO) to prepare the stock solutions. The tested compounds (**5** and **6a**–**6v**) and reference drugs were then prepared in liquid Luria–Bertani media. The required concentrations were 125, 62.5, 31.25, 15.63, 7.81, 3.91 and 1.95 μg/mL, respectively (DMSO < 0.5%). Triplicate experiments were conducted.

### 3.4. Calculation of Molecular Physicochemical Properties

Molecular properties, including the octanol–water partition coefficient, the topological polar surface area, the number of hydrogen bond acceptors, the number of hydrogen bond donors, the number of rotatable bonds, and the molecular weight, were calculated using the Molinspiration tool www.molinspiration.com.

### 3.5. Cell Toxicity Evaluation

The cytotoxicity of highly active compounds **6p** and **6t** was assessed using the 3-(4,5-dimethyl-2-thiazolyl)-2,5-diphenyl-2-*H*-tetrazolium bromide (MTT) assay, as described previously [13,14]. Baby hamster Syrian kidney cells BHK-21 were treated with compounds **6p** and **6t** at concentrations of 3.91, 7.81 and 15.62 μg/mL. After incubation, 10 μL (5 mg/mL) of MTT in phosphate-buffered saline was added to each well, and the cells were incubated in 5% CO_2_ at 37 °C for 4 h. The medium was removed. Then, the DMSO (150 *µ*L) was added to each well. The wells were shocked for 15 min in a flat shaker to dissolve purple formazan crystals. Finally, the plate was tested at 540 nm. The untreated group was considered as the control. Triplicate experiments were conducted. The data were analyzed by GraphPad Prism 6.0 (GraphPad Software, San Diego, CA, USA). The formula used for cell viability was (absorbance of compound-treated cells/absorbance of untreated cells) × 100%.

### 3.6. Germination Rate Bioassay

The seeds of two herbaceous plants, turnip (*Raphanus sativus*) and wheat (*Triticum aestivum* L.), were used for the bioassay. The procedure was conducted according to the reported protocol [14]. The plant seeds were washed with running water for 2 h, soaked in 0.3% KMnO_4_ for 15 min, and flushed until they were colorless. The compounds **6p** and **6t**, positive control and blank solvent acetone, were added to 12-well plates with filter paper to final concentration of 3.91 μg/mL. After the evaporation of acetone, the plant seeds were sown in the microdishes of 12-well plates and irrigated with sterile deionized water. Triplicate experiments were conducted. The plates were then incubated at 26 °C for 48 h, and the germination rates were calculated according to Equation (1). Germination rates were expressed as means ± the standard deviation (SD) for three replicates.

Germination rate (%) = number of germinated seeds/total number of seeds(1)

## 4. Conclusions

In this study, twenty-two new quaternarization 10-methoxycanthin-6-one derivatives were designed and synthesized based on our previous work. Their antibacterial activity was evaluated against three bacterial strains including two kinds of agricultural pathogenic bacteria. Four compounds (**6f**, **6i**, **6p** and **6t**) displayed 8-fold superiority (MIC = 3.91 µg/mL) against *R. solanacearum* and *P. syringae* compared to agrochemical streptomycin sulfate. Simultaneously, the structure–activity relationships were summarized, which provided some important guidance for the development of antibacterial agents. The good “drug-like” properties, low cytotoxicity and no inhibition on seed germination of the highly active compounds **6p** and **6t** further confirmed their potential as bactericides and preservatives, which is of great meaning both to agricultural production and food industry.

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
