# Peer review of "Synthesis and In Vitro Antibacterial Activity of Quaternized 10-Methoxycanthin-6-one Derivatives"

_molecules, 2019, doi:10.3390/molecules24081553_

Round 1
Reviewer 1 Report
Well designed and well presented experiments. English should be polished (eg. x and y are major coméponenst of plant pathogens; three kinds of bacteria; conformed instead of confirmed), apart from this the paper is excellent.
Author Response
Well designed and well presented experiments. English should be polished (eg. x and y are major coméponenst of plant pathogens; three kinds of bacteria; conformed instead of confirmed), apart from this the paper is excellent.
Response: We have corrected it. Question 1 (Lines 36−37); Question 2 (Line 81−82); Question 3 (Line 73).
Reviewer 2 Report
The paper demonstrates a nice process where quaternisation of a molecule can alter and enhance its antibacterial activity against three very prominent bacterial strains. The article is interesting and for the most part well written and structured, consequently, I would recommend publication following some minor edits highlighted below and in the attached manuscript.
· I would make some amendments to the abstract as this could be somewhat clearer and more ‘punchy’, giving the reader a greater impression of the results and less emphasis on the background.
· There is little evidence for the purity of the compounds produced. Moreover, there is little discussion about the compound identity in the text. I would, therefore, like to see a supporting information document containing the 1H and 13C NMR data for all the synthesised compounds.
· There is red text throughout the document, please remove this
· The use of the word ‘quaternization’ is incorrect in nearly all the places that it appears, it should read ‘quaternized’. Please make these amendments to improve the grammar of your document.
· Finally, to aid visualisation of the antibacterial data against the compounds it would be beneficial to combine the table at the bottom of scheme 1 with biological results table 1. This would also allow the removal of the box around scheme 1 which looks odd.
Author Response
The paper demonstrates a nice process …… I would recommend publication following some minor edits highlighted below and in the attached manuscript.
1. I would make some amendments to the abstract as this could be somewhat clearer and more ‘punchy’, giving the reader a greater impression of the results and less emphasis on the background.
Response: We have re-written the Abstract according to the suggestion (Lines 11−13).
2. There is little evidence for the purity of the compounds produced. Moreover, there is little discussion about the compound identity in the text. I would, therefore, like to see a supporting information document containing the 1H and 13C NMR data for all the synthesised compounds.
Response: The purity of the compounds has been proved using HPLC (Lines 65−66; Supporting Information). We have added some discussion about the compound identity in Lines 70−71. Moreover, the supporting information containing the NMR and HRMS spectra, were also submitted.
3. There is red text throughout the document, please remove this.
Response: We have removed them.
4. The use of the word ‘quaternization’ is incorrect in nearly all the places that it appears, it should read ‘quaternized’. Please make these amendments to improve the grammar of your document.
Response: We have corrected it in the appropriate places including the subject.
5. Finally, to aid visualisation of the antibacterial data against the compounds it would be beneficial to combine the table at the bottom of scheme 1 with biological results table 1. This would also allow the removal of the box around scheme 1 which looks odd.
Response: We appreciate you very much for your constructive suggestions. We have added the structures of the substitutions in Scheme 1. Moreover, we added the substitutions of the compounds in Table 1, which could enhance the readability of the manuscript. In addition, we removed the box in Scheme 1.

This manuscript is a resubmission of an earlier submission. The following is a list of the peer review reports and author responses from that submission.
Round 1
Reviewer 1 Report
The authors present an investigation on the antibacterial activity of quaternary, N-benzylated 10-methoxycanthin-6-ones. This manuscript is an uninspired continuation of previously published work (ref. 12) on very similar compounds (only the 10-methoxy substituent was absent in the previous paper). In my opinion, this represents an undue fractionation of results.
Further, it is not obvious to me why the authors performed almost the same kind of investigations with 10-methoxycanthin-6-one – Figure 1 demonstrates that this compound has not at all advantages over the previously investigate unsubstituted canthin-6-one compound 1).
Response: We have synthesized a series of 3-N-substituted canthin-6-ones in 2015. We found that quaternization modification could significantly improve the antibacterial activity (Bioorg. Med. Chem. Lett. 2016, 26, 580). In 2016, we synthesized natural product 10-methoxycanthin-6-one and evaluated its antibacterial activity against B. cereus, B. subtilis, R. solanacearum, P. syringae (Molecules 2016, 21, 390). We found that 10-methoxycanthin-6-one showed 2-fold superiority against B. cereus (MIC = 3.91 μg/mL) than canthin-6-one 1 (MIC = 7.81 μg/mL). Moreover, 10-methoxycanthin-6-one per se displayed 4-fold superiority (MIC = 7.81 μg/mL) against plant pathogen P. syringae and 2-fold superiority (MIC = 15.63 μg/mL) against plant pathogen R. solanacearum compared with agrochemical streptomycin sulfate, respectively. So we deduced that quaternization of 10-methoxycanthin-6-one might bring more antibacterial candidates. Although it shows no advantage against R. solanacearum compared with canthin-6-one 1, chemical diversity of 10-methoxycanthin-6-one may surprise us. Eight compounds (6e, 6f, 6i, 6k, 6l, 6m, 6p and 6t) displayed 4-fold superiority (MIC = 3.91 μg/mL) against R. solanacearum than lead compound 5, 2-fold superiority than canthin-6-one 1 and 8-fold superiority than streptomycin sulfate.
And inexplicably the design of the present study deviates from the previously published one in some aspects: the authors use some different bacterial strains here (in my opinion 3 strains are not enough), they omitted investigation of N-alkyl residues, and they used different reference antibiotics (the prominent ampicillin is missing here). Further, the significant activity of reference antibiotic fosfomycin (1.95 microgram/mL is reported in ref. 12) is missing in this manuscript.
Response: Natural-sourced antibacterial agents for food preservations and agrochemical bactericides have received much attention because of their low toxicity, low residue and environmental friendliness. Based on our previous study (Bioorg. Med. Chem. Lett. 2016, 26, 580; Molecules 2016, 21, 390), we selected these three bacterial strains which are more valuable from the perspective of agricultural production and food industry.
As we reported in 2016, N-alkyl quaternization contributed little to activity improvement with lower synthesis yield. So we abandoned it.
We have added the activity data of broad spectrum antibiotic fosfomycin sodium and ampicillin sodium, and agrochemical bactericide streptomycin sulfate.
For R. solanacearum, the activity of fosfomycin is 1.95 μg/mL (Bioorg. Med. Chem. Lett. 2016). However, agricultural bactericide streptomycin sulfate was applied in the actual production. So we selected streptomycin sulfate as main positive control against R. solanacearum and P. syringae.
Figure 1 is not informative, since part of the MIC values are given in microgram/mL units, others in mmol/mL.
Response: We have corrected it.
Conversion of a neutral compound into a quaternary salt will change pharmacokinetic behavior significantly. The authors did not at all discuss this topic.
Response: We are very grateful for your kind comments. We have calculated the physicochemical properties of 10-methoxycanthin-6-one 5 and the highly active derivatives 6p and 6t. In addition, we added the cytotoxicity and germination rate evaluation of compounds 6p and 6t, which further proved the potential and application value of this class of compounds.
The novelty of the results presented here is poor, and the design of the investigation is defective.
My recommendation: reject.
Reviewer 2 Report
In the manuscript “Synthesis and in vitro Antibacterial Activity of Quaternization 10-Methoxycanthin-6-one Derivatives”, the authors propose to generate bioactive compounds by quaternization reaction of 10-methoxycanthin-6-one with superior antibacterial activity. The paper has interest and falls in the scope of the journal. Nevertheless, there are some minor comments that could be addressed. I have some specific comments that may be useful.
Specific comments:
1. Abstract needs major revision. Please include objective, experimental approach, major results and conclusions.
Response: We have re-written it (Lines 11−25).
2. It would be important to discuss some issues such as: the concentrations used in the studies of antimicrobial tests have any relevance? What is the potential use of these compounds? Please include some of these issues in the discussion.
Response: We are very grateful for your kind comments. The concentration gradient was set according to our previous work (Bioorg. Med. Chem. Lett. 2016, 26, 580; Molecules 2016, 21, 390) (Lines 81−82). In order to highlight the potential of these compounds, we calculated the molecular physicochemical properties, and added the cytotoxicity and germination rate experiments. In addition, we have re-written the discussion section (Lines 346−355).
Reviewer 3 Report
The data presented show the antibacterial activity of 22 derivatives of 10-methoxy canthin-6-one alkaloids against 3 species of bacteria.
Comments
Paragraphs 1 and 2 of the Introduction need to be rewritten because they do not provide the reader with suitable background information on the rationale for the presented research and are unfocused..
For example, paragraph one provides general comment on the serious nature of antibiotic resistance and the need for new drugs. The authors then mention 3 species of bacteria with no explanation why these 3 were selected. In the context of global antibiotic resistance these three species are not of great concern. If R. solanacearum and P. syringae are of concern in regard to crop infections then this should be the focus of their introduction not general comments re antibiotic resistance. The choice of B. cereus is strange - yes the organism does cause food poisoning but antibiotic resistance is not problem with this pathogen in human medicine.
Paragraph 2 has little relevance to the subsequent presented work.
Paragraph 3 is good and focused on providing information on previous data that underpins the present study.
Response: We are very grateful for your kind comments, and we have re-written the introduction.
Figure 1 - MIC values of 1, 2 and 5 are given as ug/mL but 3 and 4 are nmol/mL which is incorrect.
Response: We have corrected it.
Scheme 1 - compounds 7-10 are not named. The compound numbering is confusing and not sequential eg. 7, 8, 9 and 10 are shown before 5 and 6a-6v.
Response: We have named the compounds 7−10 in lines 60−65 (Chemistry section). Compounds 5 and 6 were introduced in Figure 1. So they appear in front of compounds 7−10 based on the principle that only one name of the same compound.
No explanation is given to the reader what the significance of 'X' is?
Response: We have explained it in Figure 1.
Table 1 shoes the MIC data for the 22 derivatives. Why are the MIC values such strange values? In the Methods the authors state that they used the NCCLS method where MIC values are usually expressed as 512, 256, 128, 64, 32, 16, 8, 4, 2, 1, 05, 0.25 etc. mg/L not 15.63, 7.81, 3.91?
Response: The concentration gradient was set according to our previous work (Bioorg. Med. Chem. Lett. 2016, 26, 580; Molecules 2016, 21, 390) (Lines 81−82). So the MIC values were determined as described by the National Committee for Clinical Laboratory Standards with minor modification (Molecules 2016, 21, 390; J. Med. Chem. 2014, 57, 8398; J. Med. Chem. 2017, 60, 8456) (Lines 307−308).
line 73 - the authors use fosfomycin sodium and propineb as positive controls but provide insufficient detail on why these were chosen? Are they used to treat crop infections by the bacterial species they use in the study? Do these species become resistant to these compounds?
This lack of clarity on the rationale for their selection goes back to previous comments re the content of the Introduction.
Response: We are very grateful for your kind comments. The introduction was re-written by us. Bacillus cereus is pathogenic and is the common cause of spoilage of milk, cream and cooked rice. When people eat B. cereus contaminated food, they will suffer from food poisoning with these symptoms nausea, vomiting and diarrhea. Ralstonia solanacearum and Pseudomonas syringae are major components of plant pathogens, which could infect the plant tissues during the whole plant growth cycle. Natural-sourced antibacterial agents for food preservations and agrochemical bactericides have received much attention because of their low toxicity, low residue and environmental friendliness. Although propineb displayed good antibacterial activity, it is often used as the positive control for antifungal activity. Here, broad spectrum antibiotic fosfomycin sodium and ampicillin sodium, and agrochemical bactericide streptomycin sulfate were used as the positive controls (Food Chem. 2018, 253, 211; Eur. J. Med. Chem. 2018, 143, 558) (Lines 82−83).
The significance of the results needs to be studied carefully. The differences in antibacterial activity between for example 6p and propineb are only 2-fold. In terms of MIC values this is not a huge increase in antibacterial activity. In fact, is it reproducible as no indication of replicate measurements is given? So conclusions like on line 95 are overstated.
Response: Thanks for your kind comments. We have re-written the results and conclusions (Lines 346−355). Triplicate experiments were conducted (Line 317).
The section on structure-activity relationships requires more detail. What is the significance of their findings in terms of the applicability, usefulness, toxicity of these compounds. More detailed explanation is required. The authors could improve their study by including some data/experimentation on the toxicity of these compounds versus RBCs and cultured mammalian cells for example.
Response: We have added more details in structure-activity relationships (Lines 101−103, 105−107, 110−111).
In order to highlight the potential of these compounds, we calculated the molecular physicochemical properties, and added the toxicity and germination rate experiments.
Their physicochemical data conform to the Lipinski rule which demonstrate they may have good “drug-like” properties and bioavailability (Lines 112−127).
The cell viabilities of the highly active compounds 6p and 6t were more than 96% on BHK-21 cells even up to concentrations of 15.62 μg/mL (2-fold or 4-fold MIC values). The results suggest that these molecules are of low toxicity against mammalian cells (Lines 128−137).
The disease caused by R. solanacearum could be controlled before planting. So the effect of bactericides on seed germination rate is more important. Compounds 6p and 6t showed no inhibition on germination rates under an effective antibacterial dose
(3.91 μg/mL), which indicated that these molecules would not inhibit plant growth when it acts as bactericides (Lines 138−145).
These results confirmed the potential of quaternarization 10-methoxycanthin-6-one derivatives as bactericides and preservatives, which is of great meaning both to agricultural production and food industry.